# The Potential for Effect of a Six-Week Training Program for Gait Aid Use in Older People with Dementia with Unsteadiness of Gait: A Pilot Study

**DOI:** 10.3390/jcm12041574

**Published:** 2023-02-16

**Authors:** Den-Ching A. Lee, Elissa Burton, Claudia Meyer, Terry P. Haines, Susan Hunter, Helen Dawes, Plaiwan Suttanon, Stephanie Fullarton, Fiona Connelly, Julie C. Stout, Keith D. Hill

**Affiliations:** 1Rehabilitation Ageing and Independent Living (RAIL) Research Centre, Monash University (Peninsula Campus), Frankston, VIC 3199, Australia; 2National Centre for Healthy Ageing, Monash University and Peninsula Health, Frankston, VIC 3199, Australia; 3Curtin School of Allied Health, Faculty of Health Sciences, Curtin University, Perth, WA 6102, Australia; 4enAble Institute, Faculty of Health Sciences, Curtin University, Perth, WA 6102, Australia; 5Bolton Clarke Research Institute, Forest Hill VIC 3131, Australia; 6Centre for Health Communication and Participation, La Trobe University, Bundoora, VIC 3086, Australia; 7College of Nursing and Health Sciences, Flinders University, Adelaide, SA 5042, Australia; 8School of Primary and Allied Health Care, Faculty of Medicine, Nursing and Health Sciences, Monash University, Frankston, VIC 3199, Australia; 9School of Physical Therapy, University of Western Ontario, London, ON N6G 1H1, Canada; 10NIHR Exeter BRC, Medical School, University of Exeter, Exeter EX1 2LU, UK; 11Thammasat University Research Unit in Health, Physical Performance, Movement, and Quality of Life for Longevity Society, Faculty of Allied Health Sciences, Thammasat University, Pathumthani 12120, Thailand; 12Department of Geriatric Medicine, Armadale Kalamuda Group, Armadale Health Service, Perth, WA 6112, Australia; 13Turner Institute for Brain and Mental Health, School of Psychological Sciences, Monash University, Melbourne, VIC 3800, Australia

**Keywords:** gait aid, dementia, training, spatiotemporal, gait, falls, physiotherapist, safety

## Abstract

This study examined the potential for effect of a six-week gait aid training program for people with dementia on spatiotemporal gait outcomes, perception of use, and falls with gait aid use. The program utilised four 30-min physiotherapy home visits, scheduled at weeks 1/2/3/6, and was enhanced by carer-supervised practice. Falls and the physiotherapist’s clinical judgement of participants achieving safe gait aid use during and after the program were described. Perception ratings at each visit were measured using Likert scales which, along with the spatiotemporal outcomes using the gait aid (Time-Up-and-Go-Test, 4-m-walk-test, Figure-of-8-Walk-Test with/without a cognitive task) at weeks 1 and 6, and at weeks 6 and 12 (6-week post-program), were examined with ordinal logistic regression analyses. Twenty-four community-dwelling older people with dementia and their carers participated. Twenty-one (87.5%) older people achieved safe gait aid use. Twenty falls occurred, and only one faller was using their gait aid when they fell. Walking speed, step length, and cadence significantly improved when walking with the gait aid at week 6 compared with week 1. No significant improvements in spatiotemporal outcomes were retained at week 12. Physiotherapists were more likely to agree that gait aid use had improved walking safety among older people with dementia with subsequent training visits. Larger studies of the gait aid training program are needed for this clinical group.

## 1. Introduction

Balance and mobility impairments are common among older people, especially among those who have dementia. Dementia impacts physical performance [1] alongside attentional and executive functioning [2], often resulting in greater levels of functional impairment, particularly when performing more cognitively demanding physical tasks such as activities of daily living in the home environment [3]. This impaired physical and cognitive functioning places individuals with dementia at higher risk of falls compared with those without dementia [4]. Approximately one-half of the people with Alzheimer’s disease and vascular dementia fall each year [4] compared with this statistic being approximately one-third in the general older population [5,6]. Early identification and intervention for physical impairments are vital for improving independence, function, and safety for people with dementia. 

Gait aid prescription, the provision and instruction on the appropriate use of gait aids such as a 4-wheeled walker or walking stick, is a standard approach for improving walking independence and mobility for older people with balance or mobility impairments. Although gait aid use is common among older people [7], evidence shows that gait aids are not always used correctly. For example, one study identified that incorrect and potentially unsafe walker use among older people occurred in 16% of single stance phases and 29% of double stance phases of gait [8]. Additionally, a physiotherapy review found that 32% of older people who were recovering from a hip fracture had an incorrect gait aid or were incorrectly using the gait aid in the first six months after hospital discharge [9]. Indeed, this is not surprising considering that a survey showed that 37% of community-dwelling older people with dementia did not receive advice from a health professional regarding gait aid use when they obtained their gait aids [10]. A lack of training may lead to the improper use of gait aids and inadequate gait patterns, resulting in a paradoxical reduction in gait stability and increasing users’ risk of falls [11]. Having an altered gait pattern such as reduced gait velocity [12], shorter step length [13], slow (decreased) cadence (steps/time) [14], and increased gait variability (e.g., stride time variability) [15] have been associated with future falls in community-dwelling older people. 

There is also a difference of opinions on whether the instruction of gait aid use is feasible for people with dementia, which is in part related to their capacity to learn new motor skills and understand instructions. A systematic review of motor skill learning in people with Alzheimer’s disease demonstrated intact implicit motor learning capabilities across all studies. Importantly, this occurred even in the presence of severe explicit memory impairment [16]. However, important factors need to be considered in facilitating the implicit learning of a motor skill (i.e., repeated running of the same motor task, which does not require an intact episodic memory), such as using a gait aid to walk in people with dementia. In particular, there is a need for constant and consistent practice that emphasises movement consistency in the same environment [17], as well as a training program that is tailored to the needs and ability of people with dementia [18]. Despite this, a recent review found that most studies of gait aid prescription for older people, including those with dementia, had implemented a single or brief gait aid instruction session [19]. Furthermore, the gait aid prescription had inconsistent effects on improving spatiotemporal gait outcomes in older people with mobility problems or fall risks, including in studies of people with Alzheimer’s disease [19,20,21]. There has been little consideration that learning and familiarity with the gait aid in people with dementia may take several sessions to achieve effective outcomes or of the form of learning that may be more effective. Impaired sensorimotor circuits affecting the ability to learn and perform new motor patterns include reduced working memory combined with a decreased capacity for implicit learning and reduced intero- and exteroceptive feedback affecting error detection, which have an impact on the speed at which people with dementia learn new motor activities [22]. However, people with dementia retain such implicit motor learning capabilities and an ability to learn novel functional tasks despite explicit memory changes [16]. 

Currently, the benefits of providing a gait aid and using an extended training program to teach the safe and correct use of gait aids for people with dementia is unknown. This pilot study examined the potential effects of providing a gait aid, used an extended training program to teach the safe and correct use of gait aids, and observed spatiotemporal gait outcomes, perception of safety, and appropriateness of gait aid use in people with dementia. It also investigated the success of the training program based on the clinical judgement of study physiotherapists, gait aid usage, and falls with the gait aid use during and after the training program.

## 2. Materials and Methods

### 2.1. Design

A pre-post single pilot group design using an extended gait aid training program was undertaken. The intervention was based on published evidence regarding the ineffectiveness of a single or brief gait aid instruction session for older people with mobility problems or fall risks, including in people with Alzheimer’s disease, on spatiotemporal gait outcomes, safety, and adherence to gait aid use [19,20,21]. 

### 2.2. Participants

Participants comprised people who were: (1) aged ≥65 years; (2) living in the metropolitan area of Melbourne and Perth, Australia; (3) medically diagnosed with dementia by a healthcare practitioner (e.g., a geriatrician or general practitioner), assessed as having cognitive impairment (Rowland Universal Dementia Assessment Scale (RUDAS) score of ≤22) [23], or those who were receiving a dementia and cognition supplement for homecare; (4) unsteady when walking or turning (self or informal carer’s report) or had fallen in the preceding 12 months; (5) willing to learn the use of a gait aid or requiring a more supportive type of gait aid that was different from what they were currently using; and (6) having a consenting person responsible and/or an informal carer who was able to supervise practice and support safe gait aid use as instructed by the study physiotherapists. 

For the inclusion criteria, a dementia and cognition supplement is an additional fund provided by the Australian Commonwealth Government that helps homecare providers with the cost of caring for people who have moderate to severe cognitive impairment [24]. A fall was defined as “inadvertently coming to rest on the ground, floor or other lower level, excluding intentional change in position to rest in furniture, wall or other objects” [25].

Exclusion criteria captured people who: (1) required manual assistance for walking (with or without a gait aid); (2) were not able to walk, e.g., bed- or wheelchair-bound; or (3) planned to be away from home during the six-week training program (including their informal carers).

People with dementia or cognitive impairment and their carers were recruited through a carer’s survey conducted earlier by the research team, general practitioner clinics, community care organisations, Dementia Australia’s website, Dementia Downunder’s Facebook page, paid Facebook and Probus e-newsletter advertisements, local city councils, dementia day respite centres, forget-me-not cafes, and community radio advertisements.

Physiotherapists with extensive experience working with older people with dementia or cognitive impairment and carers were employed in this study to deliver the gait aid training intervention.

This study was approved by the Human Research Ethics Committees of Monash University (ID 25104), Curtin University (ID HRE2020-0614), and Bolton Clarke (ID 223). Consent was obtained from all participants and, if required, from their person responsible (for the person with dementia) prior to assessment.

A total of 59 potential participants were assessed for eligibility. Of these, 35 were excluded. Participant flow through the study is shown in Figure 1.

Twenty-four older people were recruited and commenced gait aid training intervention (Table 1). Among the sample, 23 had diagnosed dementia and one had diagnosed cognitive impairment (based on RUDAS, but no formal diagnosis). Ten (41.7%) of the older people were female, and the mean age was 81.9 years (SD 5.5). The mean RUDAS score was 16.5 (SD 7.3), with 11 (45.8%) being classified as having mild dementia (RUDAS scores 17–22), 9 (37.5%) as having moderate dementia (RUDAS scores 10–16), and 4 (16.7%) as having severe dementia (RUDAS scores < 10). Alzheimer’s disease was the most common dementia type, accounting for 41.7% of all participants. The majority had difficulty moving around (79.2%) and outside of their home (91.7%), negotiating stairs (95.8%), and bending to pick up an object from the floor (58.3%). Twenty-one (87.5%) had at least one fall in the past year.

### 2.3. Assessment

Participants and their person responsible and/or informal carer were interviewed by study physiotherapists at home for demographic information, presence of formal diagnosis of dementia and/or other medical conditions, RUDAS cognitive score, and self or carer-reports on: (1) steadiness in walking, turning, and negotiating stairs, (2) ability to remember new information, and (3) falls in the last 12 months. The participants’ physical capacity (ability to ambulate inside and outside their home, walk up and down three stairs without a handrail, bend and pick up an object from the floor) was observed by the study physiotherapists. All study physiotherapists were provided with the URL links to the online surveys, which were developed by the research team using the Qualtrics survey platform to guide interaction flow with participants and for online data entry after each home visit (see Appendix A).

#### 2.3.1. Mobility Assessments

All mobility assessments were performed in the participant’s home by study physiotherapists who were experienced in aged care at week 1 (first training session) and week 6 (fourth and last training session) using the recommended gait aid. Spatiotemporal gait measures were assessed using a stopwatch for the 4-m-walk-test to calculate the walking speed (distance/time), step length (distance/number of steps taken), and cadence (steps/time). Functional mobility was assessed using the Time-Up-and-Go Test [26]. The ability to walk on a curved path was measured using the Figure-of-8-Walk Test [27] with and without a concurrent cognitive task (counting backwards by ones from fifty while walking).

#### 2.3.2. Perception Ratings of Safety and Appropriateness of Gait Aid Use at Each Training Session

Perception of safety and appropriateness of gait aid use was rated by participants, informal carers, and study physiotherapists at each training session (i.e., week 1, 2, 3, 6) using a Likert scale of “Strongly agree” (coded as 1), “somewhat agree” (coded as 2), “neither agree nor disagree” (coded as 3), “somewhat disagree” (coded as 4), and “strongly disagree” (coded as 5) to the statements: (a) the gait aid has improved the participant’s steadiness in walking, (b) the gait aid has improved the participant’s safety in walking, and (c) using the gait aid is appropriate for the participant.

### 2.4. Intervention

The intervention consisted of a home-based, carer-enhanced, and individually tailored training program for safe gait aid use, which was delivered over six weeks. All study physiotherapists attended a three-hour training course delivered by the chief investigator and received a manual on the assessment protocol and principles of motor skill training based on errorless learning [22] to help participants achieve implicit learning for gait aid use (see Appendix A). Each study physiotherapist visited their participants at their home four times, scheduled at week 1, 2, 3, and 6, to provide training for safe gait aid use. Informal carers were required to be present to observe the training sessions and followed written instructions provided by the study physiotherapists at each home visit to supervise the practice of gait aid use with the participant; the carers were also required to enable frequent and constant practice in between these scheduled visits if considered safe by the study physiotherapists. Regular contacts between the chief investigative team and the study physiotherapists were maintained to ensure compliance with the designed protocol during the study.

The first home visit consisted of an initial assessment of the participant’s mobility and balance (as described above), a discussion with the participant and informal carer about their mobility requirements and preference of gait aid type, followed by a selection of the most suitable gait aid and adjustment of the gait aid to the participant’s height. All gait aids (either a 2- or 4-wheeled walker or a single-point stick) were purchased for the participants from a mobility equipment speciality supplier in Melbourne and Perth, Australia, and provided at no cost to the participants.

Different types of gait aids were recommended and provided by the study physiotherapists for the training program. Fifteen participants (62.5%) were provided with a 4-wheeled walker, six (25%) with a single point stick, two (8.3%) with both a 4-wheeled walker and a single-point stick, and one (4.2%) with a 2-wheeled walker. Twenty participants (83.3%) changed from not using a gait aid to walking with the provided gait aid. Four (16.7%) changed from using a less supportive gait aid to using a 4-wheeled walker (Table 1).

The training program allowed personalised variation as deemed appropriate by the study physiotherapists based on the mobility requirements (e.g., indoors, outdoors, step/kerb/stairs/ramp) and learning capacity of each participant. All visits typically consisted of approximately 30 min of training for safe gait aid use, focussing on the techniques of safe gait aid use and gait patterns for the participant’s mobility requirements.

### 2.5. Gait Aid Usage and Falls during the Training Program

Informal carers recorded the supervised gait aid practice in diaries weekly over the six week duration. They reported adherence to gait aid practice, which was compared against the study physiotherapist’s written instructions that were given out at each visit during the next scheduled visit. Adherence was self-reported based on the question, “*Did the person with dementia or cognitive impairment and informal carer adhere to home practice in the last week as per the physiotherapist’s recommendations?*”. Informal carers selected one of four responses: *(a) “Yes. Full adherence to the frequency and duration of practice”, (b) “No. Partial adherence to the frequency and duration of practice”, (c) “No practice at all”, or (d) “No home practice was recommended for the last week”*. 

Falls (as defined above) were also recorded using diaries over the six-week training program. Participants and/or informal carers reported if any falls had occurred at each week during the next scheduled visit, if the participants were using their gait aid when they fell (*Yes, No, or No falls*), and the circumstance of the fall(s).

### 2.6. Program Success and Strategies Used by Study Physiotherapists to Train Safe Gait Aid Use

The study physiotherapists evaluated whether participants had successfully demonstrated the safe use of the gait aid in each training session at week 1, 2, 3, and 6 using their clinical judgement (*Yes, Partially, or No*)*,* and whether the participants had successfully achieved safe gait aid use in the overall training program using their clinical judgement (*Yes or No*).

The study physiotherapists reported strategies that they used to assist participants learn safe gait aid use during the training session. They selected one or more from eight strategies: *(a) “constant pattern of practice”, (b) “constant pattern of instruction”, (c) “memory aid”, (d) “verbal cues”, (e) “visual cues”, (f) “start with non-complex environment, then progress to more complex environment”, (g) “avoid concurrent tasking during training session” (e.g., minimise talking during task performance) and (h) “other” with an open text response.*

### 2.7. Follow-Up Post-Training Program

Participants whom the study physiotherapists recommended, using their clinical judgement, to continue using the gait aid after the training program were followed up at week 12 (i.e., 6 weeks post-program). Self-reported gait aid usage was assessed through telephone interviews (*Yes: at all times when walking; partial use: only for certain tasks or at times; or no: discontinued using the gait aid completely*). An additional home visit was made to repeat the mobility assessments for those who had continued to use the recommended gait aid at all times or some of the time. Spatiotemporal gait measures were repeated using the 4 m walk test. Functional mobility was re-assessed using the Time-Up-and-Go Test. The ability to walk on a curved path was re-measured using the Figure-of-8-Walk Test with and without a concurrent cognitive task (counting backwards by ones from fifty while walking).

Falls were also recorded using diaries over the six-week follow-up period. Participants and/or informal carers reported the number of falls that occurred in the last 6 weeks at week 12, if the participants were using their gait aid when they fell (*Yes, No, or No falls*), and the circumstances of the fall(s).

### 2.8. Data Analysis

The data were analysed with STATA SE version 15.1 (StataCorp LLC, College Station, TX, USA, 2017). Univariate ordinal logistic regression analyses with data clustered by individual participant, utilising robust standard errors [28], were employed to examine the relationships between spatiotemporal gait measures, functional mobility, and Figure-of-8-Walk Test results, each as a dependent variable, and time at (1) week 1 and 6, and (2) week 6 (program completion) and week 12 (6-week follow-up post-program) as the independent variable. Participants who attempted but were physically or cognitively unable to comprehend and execute the requirements of the mobility assessments or who were cognitively unable to comprehend and provide perception ratings, or out of the physiotherapists’ concern for participant safety if doing the assessments, were allocated a score of “999” and were included in the ordinal logistic regression analyses but excluded from the calculation of mean and SD for that variable. Participants who refused to participate in the assessments at the visit (e.g., behavioural or mood issues) were treated as “missing at random” and were not included in the ordinal logistic regression analyses. The same analyses were performed for the perception ratings of each statement regarding the safety and appropriateness of gait aid use as the dependent variable and time at week 1, 2, 3, and 6 as the independent variable. 

Spatiotemporal gait outcomes were regarded as favourable according to the literature if walking speed was higher [12], step length was longer [13], cadence was faster [14], and Time-Up-and-Go Test time was less. The ability to walk a curved path as indicated by the Figure-of-8-Walk Test was regarded as improved if the time was reduced, path walking was accurate (i.e., completed within 0.6 m boundary by comparison with the physiotherapist’s mental map of the testing space), walk and concurrent counting task time was reduced, and path walking was accurate during the concurrent counting task.

The carers’ adherence to the supervised practice of gait aids during the six-week program, falls during the program and in the follow-up period, strategies used by study physiotherapists to train safe gait aid use, and their clinical judgement of program success were summarised using descriptive statistics. 

## 3. Results

All participants finished the training program, and 20 (83.3%) were followed-up with six-weeks post-program (Figure 1). Six study physiotherapists with aged care experience ranging from 10 to 20 years delivered the gait aid training intervention to participants and carers.

### Effects of the Intervention

**Mobility assessments.** The participants significantly improved their walking speed [Coefficient (robust 95% CI): 0.57 (0.09, 1.04)], step length [Coefficient (robust 95% CI): 0.57 (0.21, 0.92)], and cadence [Coefficient (robust 95% CI): 0.85 (0.15, 1.54)] when walking with the recommended gait aid at week 6 compared with week 1 (Table 2).

Participants showed a trend for improved functional mobility when walking with the recommended gait aid at week 6 compared with week 1. In addition, participants showed a trend towards improved ability to walk a curved path with and without a concurrent cognitive task when walking with the recommended gait aid at week 6 compared with week 1. However, both of these results were not statistically significant (Table 2). Some improvement trends were retained at week 12 when compared with week 6 but were not statistically significant. The trend for faster cadence and better functional mobility continued. The accuracy of the Figure-of-8-Walk Test showed an improvement trend at week 12 compared with week 6, which was not present when comparing week 6 with week 1 (Table 3).

**Perception ratings of safety and appropriateness of gait aid use**.The majority of participants, carers, and study physiotherapists “strongly agreed” or “somewhat agreed” that using the gait aid had improved the participant’s steadiness and safety in walking and that using the gait aid was appropriate for the participant (Table 4). The study physiotherapists were more likely to agree that gait aid use had improved walking safety among older people with dementia with subsequent training visits.

**Gait aid usage and falls during and after the training program.** The adherence rates of supervised practice with gait aids as per the study physiotherapists’ instruction over the six-week program was *“full adherence to the frequency and duration of practice”* (41.7%), *“partial adherence to frequency and duration of practice”* (54.2%), *“no practice at all”* (2.8%), and *“no home practice was recommended for the last visit”* (1.4%). 

At week 12 follow-up, 75% of the participants were using their recommended gait aid *partially* (i.e., only at times or for certain tasks), 20% were using their recommended gait aid at *all* times, and 5% had *discontinued* the use of their recommended gait aid but did not provide a reason. 

There were 15 falls that occurred during the six-week program and five falls during the six-week follow-up period post-program (weeks 7–12). Two participants had one fall, and one participant had two falls between week 1 and 2. Three participants had one fall between week 2 and 3. Six participants had one fall, and one participant had two falls between week 3 and 6. Three participants had one fall, and one participant had two falls between week 7 and 12. Of the participants who fell at the various time points, only one was using the gait aid when they fell between week 2 and 3. The carer reported that the participant was adjusting the hand brakes of the walker at the time of the fall but was not walking.

**Program success and strategies used by study physiotherapists to train safe gait aid use.** For the overall training program, based on the study physiotherapists’ clinical judgement, 21 participants (87.5%) were evaluated as having achieved safe gait aid use and three (12.5%) as not having achieved safe gait aid use. Among the successful participants, eleven had mild dementia, eight had moderate dementia, and two had severe dementia. Conversely, of those who were unsuccessful, one had moderate dementia and two had severe dementia.

For each scheduled visit, based on the study physiotherapists’ clinical judgement, more of the participants could demonstrate safe gait aid use at subsequent visits. Seven participants (29.2%) were evaluated as having demonstrated safe use of gait aid at week 1, twelve (50%) at week 2, twelve (50%) at week 3, and fifteen (62.5%) at week 6. Correspondingly, fewer participants *partially* demonstrated safe gait aid use as more visits took place. In contrast, the number of participants who were unable to demonstrate safe gait aid use with subsequent visits remained much the same (Figure 2).

Over the six-week program, all physiotherapists used multiple strategies to help participants learn the safe use of their gait aid. The most used single strategies by the physiotherapists across the four training sessions were, *“starting with non-complex environment, then progress to more complex environment”* (79 responses), *“constant pattern of practice”* (73 responses), *“Verbal cues” and “Other”* (both 68 responses), *“constant pattern of instruction”* (58 responses), *“visual cues”* (37 responses), *“avoid concurrent tasking during training session”* (33 responses), and *“memory aid”* (6 responses). The most frequent “*Other*” strategies reported as being used were, “carer education to support gait aid use”, “written gait aid user information for carers and participants”, and “instructions for operating handbrakes of 4-wheeled walkers/special environments/terrains”.

## 4. Discussion

To the best of our knowledge, this is the first study that has examined the potential effect of providing a gait aid with an extended training program to teach safe and correct gait aid use for people with dementia. The extended gait aid training resulted in benefits of improved walking speed, step length, and cadence in people with dementia using a gait aid. There were no other significant differences identified in the outcomes examined. 

Our study showed that 88% of participants with dementia had achieved safe gait aid use in the six-week training program. The use of multiple strategies to facilitate implicit motor learning in people with dementia for the goal of achieving safe and successful use of gait aids, especially regarding the most frequently used strategies such as “starting gait aid training with non-complex environment” and “using a constant pattern of practice”, may be worth considering. Falls during and after the training program mostly occurred when participants were not using their gait aids. It could be because partial adherence to gait aid use was the predominant mode of use. We did not ask the reasons why participants were not using their gait aid at all times; however, this raised the question of which situations people need to use their gait aids or do not use, as the use of gait aids may be adversely impacted by environmental factors such as a lack of appropriate indoor space to manoeuvre the gait aid or personal choice to not use the aid.

Our six-week gait aid training program resulted in improved spatiotemporal gait outcomes. These findings are consistent with a systematic review of observational studies, which reported that experienced users of a 4-wheeled walker had a better spatiotemporal gait pattern when using a 4-wheeled walker compared with first-time users of a 4-wheeled walker [29]. Several observational studies also reported that experienced users of gait aids walked faster and had more swing time and stride length compared with inexperienced users [30,31,32,33]. This suggests the utility of designing multiple training sessions to improve gait aid user experience and gait patterns, which may lead to better spatiotemporal gait outcomes. Our study found that physiotherapists used multiple strategies to facilitate the learning of gait aid use in people with dementia. Traditional teaching methods focus on techniques of verbal instructions, corrective feedback, mental practice, and discovery learning, all of which require explicit memory and the ability to detect errors in performance; thus, they are less likely to be effective for people with dementia [22]. Educating carers on how to support the safe and correct use of the gait aid was one of the “Other” strategies used by the study physiotherapists, indicating the importance of the carers’ role in compensating for the cognitive impairments and reduced learning capacity in people with dementia. Additionally, our study indicated higher success at achieving safe gait aid use for people with mild or moderate dementia, though a small number of people with severe dementia did also achieve success. Clinicians may consider the likelihood of success of conducting training for gait aid use for people with different severities of dementia.

There was a lower usage of the recommended gait aid between training sessions and after the training program. The majority of participants (54.2%) and their carers only *partially* adhered to the recommended frequency and duration of practice during the program. At the end of the six-week training program, 75% of participants were only *partially* using their recommended gait aid, either at times or for certain tasks. This finding is consistent with other studies of gait aid use, where the intermittent use and non-use of gait aids are prevalent among older adults who have gait aids [34,35]. The reduced use of gait aids could have resulted in less or non-significant improvements in some spatiotemporal gait outcomes during and after the training program.

Nineteen falls occurred when the participants were not using their gait aids compared with one fall when the participant was using the gait aid. This difference suggested that non-gait aid use might lead to a significantly higher proportion of falls than among gait aid users. This was a similar issue identified in a survey of older adults who were cognitively intact and had a history of falls, where it was reported that 75% were not using their walker or walking stick when they had fallen [35]. Strategies to further increase gait aid use for greater proportions of their daily activities (e.g., extending the training program over a longer period) may help the older people to reach the ideal point where gait aid use is more consistent. In addition, there is a need for future studies to monitor gait aid use and when falls have occurred to determine what the challenges are for using gait aids in the home, e.g., narrow doorways, difficulty opening and closing doors, difficulty to take the gait aid over a step, or remembering to use. 

Some positive effects on the spatiotemporal gait outcomes were demonstrated in our extended gait aid training program. Our study used a nominal six-week duration to train safe gait aid use with interspersed physiotherapy home visits at 1, 2, 3, and 6 weeks. However, the actual duration of the gait aid training program and the training frequency required may need to be adjusted according to the gait aid or gait aids to be learnt, and the different environments and tasks that need to be trained using the gait aid or gait aids to suit individual needs. These are important factors to consider when planning future effectiveness studies.

There are strengths and limitations that need to be considered. The participants were recruited in two states of Australia; this increased the geographical generalisability of the results. The small sample size in our pilot study lacked statistical power, and without a control group, we were unable to establish causality by isolating the effect of the extended gait aid training program. Further studies are required given that the preliminary results are promising. The clinical judgement of experienced physiotherapists was used in our study for evaluating the safety of gait aid use and success in the training program. We did not define what safety entailed in the perception ratings of gait aid use. Using a pre-set criteria of safety and success definition may help to ensure a uniform standard of evaluation, and a two-arm design with randomisation to intervention or a control group with a definitive sample size to confirm the effects of the extended gait aid training intervention versus a single training session in future studies are recommended. We did not have a standard recommendation of how much each participant should be using their gait aid. The study physiotherapists could tailor the usage recommendation to the participants considering individual circumstances, e.g., carer availability to supervise use or lifestyle factors. However, gait aid use was reported by the participants and carers relative to the individual recommendation received. Self-reported adherence to the supervised practice with gait aids as recommended by study physiotherapists and self-reported falls could be subject to inaccuracy due to the subjective reporting nature of this outcome. However, all participants were provided with a home practice diary to record the practice undertaken and a fall diary to record the occurrence of falls each week for validation by the study physiotherapists at each home visit.

## 5. Conclusions

The potential for effects was identified for some spatiotemporal gait outcomes during the training program. The high level of falls when gait aids were not used is important and suggests the need to explore when and why they occur. Further research is needed for larger studies to investigate the effectiveness and the long-term effect of extended gait aid training in older people with dementia who have mobility and balance problems.

## Figures and Tables

**Figure 1 jcm-12-01574-f001:**
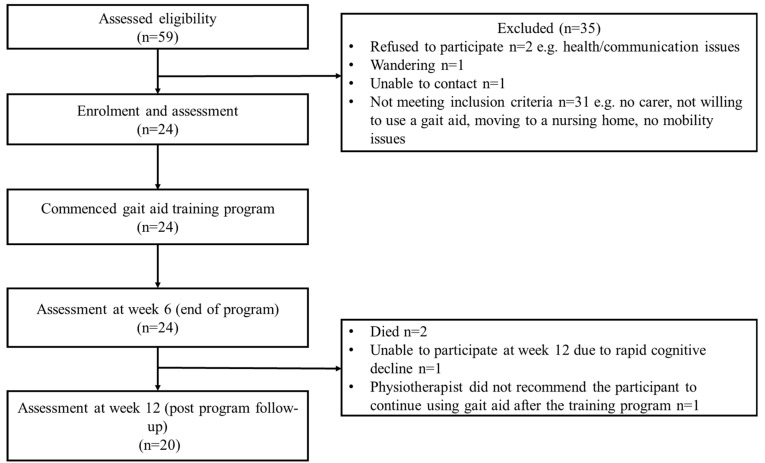
Participant flow chart.

**Figure 2 jcm-12-01574-f002:**
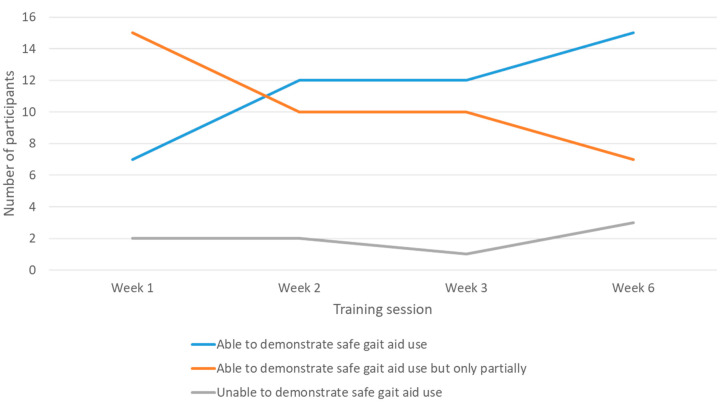
Study physiotherapists’ clinical judgement on achievement of safe gait aid use by participants at each training session.

**Table 1 jcm-12-01574-t001:** Participant demographics/characteristics at week 1.

Characteristics	All Participants with Dementia/Cognitive Impairment (N = 24)
**Age: mean (SD)**	81.8 (5.5)
**Sex: Female n (%)**	10 (41.7)
**State of residence: n (%)**	
Victoria	10 (41.7)
Western Australia	14 (58.3)
**Dementia diagnosis: n (%)**	23 (95.8)
**Type of dementia diagnosis: n (%)**	
Alzheimer’s disease	10 (41.7)
Other dementia (e.g., mixed dementia)	5 (20.8)
Lewy Body dementia	4 (16.7)
Vascular dementia	2 (8.3)
Do not know/cannot remember	2 (8.3)
Not formally diagnosed with dementia	1 (4.2)
**Years of living with dementia if diagnosed: mean (SD)**	1.0 (0.2)
**Living arrangement**	
Lives with spouse/partner	17 (70.8)
Lives alone	5 (20.8)
Lives with children	1 (4.2)
Lives with other relative(s)	1 (4.2)
**Past medical history (other than dementia/cognitive impairment): n (%); multiple choices allowed**	
Arthritis	16 (66.7)
Depression	13 (54.2)
Cancer	10 (41.7)
Anxiety	8 (33.3)
Visual impairment not correctable by glasses e.g., macular degeneration	8 (33.3)
Heart disease	7 (29.2)
Knee joint replacement	6 (25.0)
Diabetes	5 (20.8)
Parkinson’s disease	4 (16.7)
Lung disease/respiratory disease	2 (8.3)
Neurological disease (other than Stroke or Parkinson’s disease)	2 (8.3)
Stroke	1 (4.2)
Hip joint replacement	1 (4.2)
Kidney disease/renal failure	1 (4.2)
Other (e.g., osteoporosis, hearing loss, hypertension, gastric reflux, hypercholesterolemia)	15 (62.5)
**Physiotherapist report on the person’s physical capacity: n (%)**	
*Move around their home*	
Can do, appears steady	5 (20.8)
Can do but appears unsteady	18 (75.0)
Cannot do without assistance from someone else	1 (4.2)
*Move outside their home*	
Can do, reported as steady and safe	2 (8.3)
Can do but reported as unsteady or unsafe	13 (54.2)
Cannot do without assistance from someone else	9 (37.5)
*Walk up and down 3 stairs without a handrail or assistance from someone else*	
Can do without difficulty	1 (4.2)
Can do but with difficulty or needs rail assistance	15 (62.5)
Cannot do without assistance from someone else	8 (33.3)
*Able to bend and pick up an object from the floor without assistance from someone else*	
Can do without difficulty	10 (41.7)
Can do but with difficulty or unsteady	10 (41.7)
Cannot do without assistance from someone else	4 (16.7)
**RUDAS score ^a^: mean (SD)**	16.5 (7.3)
**Dementia severity ^b^: n (%)**	
Mild	11 (45.8)
Moderate	9 (37.5)
Severe	4 (16.7)
**Self (or carer’s) report on ability of the person to remember new information: n (%)**	
Can remember some information day-to-day	2 (8.3)
Can remember some information over time, but limited day-to-day	11 (45.8)
Cannot remember new information day-to-day	11 (45.8)
**Self (or carer’s) rating of the person’s steadiness in walking, turning, and negotiating stairs: n (%)**	
Mostly steady, only occasional unsteadiness in turning or going up and downstairs	4 (16.7)
Some unsteadiness in 1–2 of these activities	15 (62.5)
Unsteady in all activities	5 (20.8)
**Number of falls in past year: n (%**)	
No falls	3 (12.5)
1–3 falls	15 (62.5)
4 or more falls	6 (25.0)
**Type of gait aid recommended and provided by the physiotherapist for training: n (%)**	
4-wheeled walker	15 (62.5)
Single-point stick	6 (25.0)
4-wheeled walker (outdoors and/or long distance) and single-point stick (indoors)	2 (8.3)
2-wheeled walker	1 (4.2)
**Mobility change of participants in the training program: n (%)**	
Nil gait aid to 4-wheeled walker	11 (45.8)
Nil gait aid to single-point stick	8 (33.3)
Single-point stick to 4-wheeled walker	2 (8.3)
Nil gait aid to 2-wheeled walker	1 (4.2)
4-point stick to 4-wheeled walker	1 (4.2)
2-wheeled walker to 4-wheeled walker	1 (4.2)

^a^ RUDAS score 0–30; cut-off score 22 or less (lower score indicates greater cognitive impairment), and 23–30 is considered normal and needs to be considered in the clinical context. ^b^ Based on RUDAS score, ranges from 0–30. Score of 17–22 categorised as mild dementia, 10–16 as moderate dementia, and <10 as severe dementia.

**Table 2 jcm-12-01574-t002:** Effect of the training program for gait aid use on spatiotemporal gait outcome measures with the recommended gait aid at week 1 (first training session) and week 6 (fourth and last training session).

	Week 1 Mean (SD) or n Accurate ^a^ (%), n (Missing/Unable to Perform) and Excluded from the Calculation of Mean (SD) or n Accurate ^a^ (%)	Week 6 Mean (SD) or n Accurate ^a^ (%), n (Missing/Unable to Perform) and Excluded from the Calculation of Mean (SD) or n Accurate ^a^ (%)	Ordinal Logistic Regression Coefficient (Robust 95% CI) ^b^	*p*-Value	Direction of Coefficient Favours Intervention ^c^
* **With the recommended gait aid ^d^** *					
**Walking speed (m/s) ^e^**	0.56 (0.20), n = 0 missing/n = 0 unable	0.65 (0.23), n = 1 missing/n = 1 unable	0.57 (0.09, 1.04)	0.02 *	✓
**Step length (m) ^e^**	0.37 (0.13), n = 0 missing/n = 0 unable	0.41 (0.11), n = 1 missing/n = 1 unable	0.57 (0.21, 0.92)	<0.01 *	✓
**Cadence (steps/seconds) ^e^**	1.49 (0.29), n = 0 missing/ n = 0 unable	1.61 (0.33), n = 1 missing/n = 1 unable	0.85 (0.15, 1.54)	0.02 *	✓
**Timed-Up-and-Go Test (s) ^e^**	33.07 (21.01), n = 0 missing/n = 1 unable	29.78 (17.43), n = 1 missing/n = 1 unable	−0.15 (−0.75, 0.45)	0.63	✓
**Figure-of-8-Walk Test time (s) ^e^**	22.09 (11.91), n = 0 missing/n = 4 unable	21.97 (13.66), n = 0 missing/n = 4 unable	−0.04 (−0.45. 0.37)	0.84	✓
**Figure-of-8-Walk Test accuracy ^a^**: **Accurate n (%)**	19 (79.17%), n = 0 missing/n = 4 unable	18 (75%), n = 0 missing/n = 4 unable	0.19 (−0.56, 0.95)	0.61	✕
**Figure-of-8-Walk Test and concurrent counting task time (s):** **mean (SD) ^e^**	21.56 (9.47), n = 0 missing/n = 7 unable	23.62 (14.32), n = 0 missing/n = 6 unable	−0.10 (−0.37, 0.17)	0.49	✓ ^f^
**Figure-of-8-Walk Test and concurrent counting task accuracy ^a^:** **Accurate n (%)**	17 (70.83%), n = 0 missing/n = 7 unable	18 (75%), n = 0 missing/n = 6 unable	−0.21 (−0.63, 0.20)	0.32	✓

^a^ Accuracy of the Figure-of-8-Walk Test, coded as 1 (yes), 2 (no), and 999 (unable to perform). Accuracy was achieved if the Figure-of-8-Walk Test was completed within 0.6 m of the boundary by comparison with the physiotherapist’s mental map of the testing space. ^b^ The ordinal logistic regression analyses had included “unable to perform” data, which were coded as “999”, but excluded missing data. A positive/negative value for the coefficient for each test indicates the direction of coefficient favouring/not favouring intervention, which may be different for each spatiotemporal gait outcome measure; see below on footnote c. ^c^ Spatiotemporal gait outcomes were regarded as favourable if: walking speed was higher; step length was longer; cadence was faster, Time-Up-and-Go-Test time was less; Figure-of-8-Walk time was less; Figure-of-8-Walk was accurate; Figure-of-8-Walk Test and concurrent counting task time was less; Figure-of-8-Walk Test and concurrent counting task was accurate. ^d^ Any type of gait aid that was recommended and provided for the training program. For the two participants that used a single-point stick for indoors and 4-wheeled walker for outdoors and/or long distances, the spatiotemporal gait outcome measures were taken using the single-point stick. ^e^ Mean (SD) presented as calculated using all data, excluding those who were unable to perform the test. ^f^ There was discordance between the summative data presented in the table and the direction of the ordinal logistic regression coefficient for the Figure-of-8-Walk Test and concurrent counting task time outcome as there was one participant unable to perform this test at week 1 who improved enough to be able to undertake this test at week 6, which made the mean time at week 6 appear spuriously high relative to week 1. * Statistical significance at *p* < 0.05.

**Table 3 jcm-12-01574-t003:** Spatiotemporal gait outcome measures with the recommended gait aid at week 6 (fourth and last training session) and week 12 (post-program follow-up).

	Week 6 Mean (SD) or n Accurate ^a^ (%), n (Missing/Unable to Perform) and Excluded from the Calculation of Mean (SD) or n Accurate ^a^ (%)	Week 12 Mean (SD) or n Accurate (%), n (Missing/Unable to Perform) and Excluded from the Calculation of Mean (SD) or n Accurate ^a^ (%)	Ordinal Logistic Regression Coefficient (Robust 95% CI) ^b^	*p*-Value	Direction of Coefficient Favours Effects of Intervention after Training Program Ceased ^c^
* **With the recommended gait aid ^d^** *					
**Walking speed (m/s):** **mean (SD) ^e^**	0.65 (0.23), n = 1 missing/n = 1 unable	0.66 (0.19), n = 3 missing/n = 0 unable	−0.19 (−1.00, 0.61)	0.64	✕ ^f^
**Step length (m):** **mean (SD) ^e^**	0.41 (0.11), n = 1 missing/n = 1 unable	0.41 (0.12), n = 3 missing /n = 0 unable	−0.39 (−1.27, 0.50)	0.39	✕
**Cadence (step/seconds):** **mean (SD) ^e^**	1.61 (0.33), n = 1 missing/n = 1 unable	1.62 (0.30), n = 3 missing/n = 0 unable	0.06 (−0.81, 0.94)	0.89	✓
**Timed-Up-and-Go Test (s):** **mean (SD) ^e^**	29.78 (17.43), n = 1 missing/n = 1 unable	25.84 (8.16), n = 3 missing/n = 0 unable	−0.06 (−0.82, 0.69)	0.87	✓
**Figure-of-8-Walk Test time (s):mean (SD) ^e^**	21.97 (13.66), n = 0 missing/n = 4 unable	20.31 (8.01), n = 1 missing/n = 3 unable	0.05 (−0.78, 0.87)	0.91	✕ ^f^
**Figure-of-8-Walk Test accuracy:** **Accurate n (%) ^a^**	18 (75%), n = 0 missing/n = 4 unable	15 (75%), n = 1 missing/n = 3 unable	−0.07 (−1.50, 1.35)	0.92	✓
**Figure-of-8-Walk Test and concurrent counting task time (s):** **mean (SD) ^e^**	23.62 (14.32), n = 0 missing/n = 6 unable	21.39 (7.65), n = 1 missing/n = 5 unable	0.14 (−0.71, 0.98)	0.75	✕ ^f^
**Figure-of-8-Walk Test and concurrent counting task accuracy: Accurate n (%) ^a^**	18 (75%) n = 0 missing/n = 6 unable	13 (65%), n = 2 missing/n = 5 unable	0.28 (−0.79, 1.34)	0.51	✕

^a^ Accuracy of the Figure-of-8-Walk Test, coded as 1 (yes), 2 (no), and “999” (unable to perform). Accuracy was achieved if the Figure-of-8-Walk Test was completed within 0.6 m of the boundary by comparison with the physiotherapist’s mental map of the testing space. ^b^ The ordinal logistic regression analyses had included “unable to perform” data, which were coded as “999”, but excluded missing data. A positive/negative value for the coefficient for each test indicates the direction of coefficient favouring/not favouring intervention, which may be different for each spatiotemporal gait outcome measure; see below on footnote c. ^c^ Spatiotemporal gait outcomes were regarded as favourable if: walking speed was higher; step length was longer; cadence was faster, Time-Up-and-Go Test time was less; Figure-of-8-Walk time was less; Figure-of-8-Walk was accurate; Figure-of-8-Walk Test and concurrent counting task time was less; Figure-of-8-Walk Test and concurrent counting task was accurate. ^d^ Any type of gait aid that was recommended and provided for the training program. For the two participants that used a single-point stick for indoors and 4-wheeled walker for outdoors and/or long distances, the spatiotemporal gait outcome measures were taken using the single-point stick. ^e^ Mean (SD) presented as calculated using all data, excluding those who were unable to perform the test. ^f^ There was discordance between the summative data presented in the table and the direction of the ordinal logistic regression coefficient for walking speed, Figure-of-8-Walk Test time and Figure-of-8-Walk Test and concurrent counting task time as there was one participant unable to perform these tests at week 6 who improved enough to be able to undertake these tests at week 12, which made the mean speed and mean time at week 12 appear spuriously high or low, respectively, relative to week 6. Statistical significance at *p* < 0.05.

**Table 4 jcm-12-01574-t004:** Perception ratings by (1) participants with dementia/cognitive impairment, (2) informal carers and (3) study physiotherapists on the safety and appropriateness of gait aid use at week 1 (first training session), week 2 (second training session), week 3 (third training session), and week 6 (fourth and last training session).

Statements	Week 1	Week 2	Week 3	Week 6	Ordinal Logistic Regression Coefficient (Robust 95% CI) ^a^	*p*-Value
* **Participants with dementia/cognitive impairment** *						
**The gait aid has improved the participant’s steadiness in walking: n (%)**					−0.14 (−0.33,0.05)	0.16
**Rating scale ^b^**						
Strongly agree	10 (41.7)	12 (50)	11 (45.8)	11 (45.8)		
Somewhat agree	5 (20.8)	3 (12.5)	6 (25)	5 (20.8)		
Neither agree or disagree	5 (20.8)	4 (16.7)	3 (12.5)	2 (8.3)		
Somewhat disagree	1 (4.2)	2 (8.3)	0	0		
Strongly disagree	1 (4.2)	0	0	1 (4.2)		
Inability to rate	2 (8.3)	3 (12.5)	3 (12.5)	2 (8.3)		
Missing ^c^	0	0	1 (4.2)	3 (12.5)		
**The gait aid has improved the participant’s safety in walking: n (%)**					−0.15 (−0.37, 0.07)	0.17
Strongly agree	10 (41.7)	12 (50)	15 (62.5)	11 (45.8)		
Somewhat agree	6 (25)	6 (25)	3 (12.5)	5 (20.8)		
Neither agree or disagree	4 (16.7)	1 (4.2)	2 (8.3)	0		
Somewhat disagree	1 (4.2)	2 (8.3)	0	2 (8.3)		
Strongly disagree	1 (4.2)	0	0	1 (4.2)		
Inability to rate	2 (8.3)	3 (12.5)	3 (12.5)	2 (8.3)		
Missing ^c^	0	0	1 (4.2)	3 (12.5)		
**Using the gait aid is appropriate for the participant: n (%)**					0.02 (−0.19, 0.24)	0.84
Strongly agree	13 (54.2)	14 (58.3)	13 (54.2)	11 (45.8)		
Somewhat agree	5 (20.8)	3 (12.5)	5 (20.8)	4 (16.7)		
Neither agree or disagree	2 (8.3)	2 (8.3)	2 (8.3)	2 (8.3)		
Somewhat disagree	1 (4.2)	2 (8.3)	0	1 (4.2)		
Strongly disagree	1 (4.2)	0	0	1 (4.2)		
Inability to rate	2 (8.3)	3 (12.5)	3 (12.5)	2 (8.3)		
Missing ^c^	0	0	1 (4.2)	3 (12.5)		
* **Informal carers** *						
**The gait aid has improved the participant’s steadiness in walking: n (%)**					−0.28 (−0.62, 0.07)	0.11
Strongly agree	16 (66.7)	16 (66.7)	17 (70.8)	17 (70.8)		
Somewhat agree	8 (33.3)	7 (29.2)	5 (20.8)	2 (8.3)		
Neither agree or disagree	0	0	1 (4.2)	1(4.2)		
Somewhat disagree	0	1 (4.2)	0	0		
Strongly disagree	0	0	0	0		
Inability to rate	0	0	0	0		
Missing ^c^	0	0	1 (4.2)	4 (16.7)		
**The gait aid has improved the participant’s safety in walking: n (%)**					−0.39 (−0.85, 0.06)	0.09
Strongly agree	18 (75)	17 (70.8)	20 (83.3)	18 (75)		
Somewhat agree	5 (20.8)	5 (20.8)	2 (8.3)	2 (8.3)		
Neither agree or disagree	1 (4.2)	2 (8.3)	1 (4.2)	0		
Somewhat disagree	0	0	0	0		
Strongly disagree	0	0	0	0		
Inability to rate	0	0	0	0		
Missing ^c^	0	0	1 (4.2)	4 (16.7)		
**Using the gait aid is appropriate for the participant: n (%)**					−0.21 (−0.56, 0.14)	0.25
Strongly agree	20 (83.3)	20 (83.3)	18 (75)	19 (79.2)		
Somewhat agree	3 (12.5)	3 (12.5)	4 (16.7)	0		
Neither agree or disagree	0	1 (4.2)	1 (4.2)	1 (4.2)		
Somewhat disagree	0	0	0	0		
Strongly disagree	1 (4.2)	0	0	0		
Inability to rate	0	0	0	0		
Missing ^c^	0	0	1 (4.2)	4 (16.7)		
* **Study physiotherapists** *						
**The gait aid has improved the participant’s steadiness in walking: n (%)**					−0.23 (−0.56, 0.10)	0.17
Strongly agree	15 (62.5)	17 (70.8)	18 (75)	16 (66.7)		
Somewhat agree	8 (33.3)	6 (25)	4 (16.7)	4 (16.7)		
Neither agree or disagree	0	1 (4.2)	1 (4.2)	1 (4.2)		
Somewhat disagree	1 (4.2)	0	0	0		
Strongly disagree	0	0	0	0		
Inability to rate	0	0	0	0		
Missing ^c^	0	0	1 (4.2)	3 (12.5)		
**The gait aid has improved the participant’s safety in walking: n (%)**					−0.52 (−0.87, −0.16)	<0.01 *
Strongly agree	13 (54.2)	14 (58.3)	17 (70.8)	18 (75)		
Somewhat agree	10 (41.7)	9 (37.5)	5 (20.8)	3 (12.5)		
Neither agree or disagree	1 (4.2)	1 (4.2)	1 (4.2)	0		
Somewhat disagree	0	0	0	0		
Strongly disagree	0	0	0	0		
Inability to rate	0	0	0	0		
Missing ^c^	0	0	1 (4.2)	3 (12.5)		
**Using the gait aid is appropriate for the participant: n (%)**					−0.24 (−0.54, 0.07)	0.13
Strongly agree	14 (58.3)	16 (66.7)	17 (70.8)	15 (62.5)		
Somewhat agree	9 (37.5)	8 (33.3)	6 (25)	6 (25)		
Neither agree or disagree	0	0	0	0		
Somewhat disagree	1 (4.2)	0	0	0		
Strongly disagree	0	0	0	0		
Inability to rate	0	0	0	0		
Missing ^c^	0	0	1 (4.2)	3 (12.5)		

^a^ A negative and significant value for the coefficient for each statement indicates that participants/informal carers/physiotherapists were more likely to agree that using the recommended gait aid had improved the safety/steadiness or was appropriate for the participants in the training sessions over time. ^b^ Rating scale coded as 1 (strongly agree), 2 (somewhat agree), 3 (Neither agree nor disagree), 4 (somewhat disagree), 5 (strongly disagree), and inability to rate “999”. ^c^ Missing at random, e.g., participant refused to provide perception rating, carer left/was unexpectedly absent during the training session, or physiotherapists did not enter the data. * Statistical significance at *p*-value < 0.05.

## Data Availability

Data are available from the corresponding authors upon reasonable request.

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
