# Peer review of "The Potential for Effect of a Six-Week Training Program for Gait Aid Use in Older People with Dementia with Unsteadiness of Gait: A Pilot Study"

_jcm, 2023, doi:10.3390/jcm12041574_

Round 1

Reviewer 1 Report (Previous Reviewer 1)

Thank you for addressing the comments. 

Reviewer 2 Report (Previous Reviewer 2)

The authors have successfully responded to all my concerns regarding the manuscript, and I do not have more comments. Thank you

Reviewer 3 Report (Previous Reviewer 3)

N/A

This manuscript is a resubmission of an earlier submission. The following is a list of the peer review reports and author responses from that submission.

Round 1

Reviewer 1 Report

The study attempted to investigate the Potential Effect of a Six-week Training Program for Gait aid use in Older People with Dementia with Unsteadiness of Gait: A Pilot Study using pre-post test design. The study could contribute to the state of knowledge; however, I would like authors to clarify the following four points. 1.     It appears that the authors were involved in the implementation of the intervention; what measures did they take to ensure that their study was free of bias. 2.     Any measure that ensures compliance with the designed protocol 3.     Self-reported adherence should be reported as part of the limitations, as we are not 100% sure that they are truly adhering to the home practices recommended by the therapist. 4.     It seems that the manual was developed by the researchers and not published; was it validated? If not, it should be add to the limitations of the study.   5.     Were the authors involved in data collection/delivering the interventions/outcome assessment? 6.     Authors may wish to recommend further studies using two-arms to confirm the effects of the designed interventions.

Reviewer 2 Report

In this study, the authors examined the potential for effect of providing a gait aid with a six-week training program to teach safe and correct gait aid use in people with dementia. The experiment is well discussed, and the analysis is clear. However, there are some points that the authors need to provide and/or explain and some issues that need to be addresses. Please see attached.

Reviewer 3 Report

Reviewer’s comments for the submitted article titled “The potential for effect of a six-week training program for gait aid use in older people with dementia with unsteadiness of gait: A pilot study”

Dear Editor, Dear authors

Thank you for giving me the opportunity to review the manuscript “The potential for effect of a six-week training program for gait aid use in older people with dementia with unsteadiness of gait: A pilot study”. The authors of the submitted work aimed to investigate the results of a 6-week program of training patients with dementia in the use of walking aids regarding certain spatiotemporal parameters of walking, the perception of safety and appropriateness of gait aid use at each training session, and the adherence to gait aid usage and falls during the training period as well as 6 weeks post-training. The introduction of the study briefly and clearly highlighted the findings published so far on the questions under investigation, setting clear objectives. The Methodology was formulated with all the necessary details to enable its reproducibility and the Results provided the information produced, which eventually was adequately discussed in the Discussion section. However, while the study met the standards that were set, some queries need to be answered before deciding to publish the manuscript.

1.               A major concern regarding the current study was the lack of a control group. The authors utilized a one-group, pre-post design based on which they found some promising findings regarding the outcomes of a 6-week training program of gait aid use in patients with dementia. However, the lack of a control group (e.g., patients trained in one visit and re-evaluated at six and 12 weeks) on which the same training approach could be used as with the experimental group, leaves unanswered the question of whether the proposed training duration was really effective. Although the ineffectiveness of a single session is documented based on published evidence, the authors should consider including information regarding this issue in the Materials and Methods section to justify their experimental approach and in the limitation of the study in the Discussion section.

2.               Further to the previous comment, the authors should justify the rationale for the 6-week program for the training on the use of walking aids. Why not more or fewer weeks of training?  Also, why the visits occurred in the 1st, 2nd, 3rd, and 6th weeks and not spread out more evenly?

3.               Please define whether the participants in this study were selected from the urban areas of Perth and Melbourne. If yes, how do they differ from each other, other than geographic area, so that they are considered to contribute to increasing the generalizability of the study results?

Minor comments

·       Page 3: Please replace the subtitle “Methods and Methods” with “Materials and Methods”

·       Page 3, 3rd paragraph of the Materials and Methods: Please use appropriate numbering … (2) in missing

·       Tables 2 and 3: Please consider modifying Tables by deleting the unnecessary text from the cells. For example, it seems unnecessary to repeat that (n) participants were "... unable excluded from this calculation" on each cell since it is explained in the second column title. Was the slash used in the third column as a substitute for “or”? In this case, there was “…1 missing…” and “…1 unable excluded from this calculation…”, which means two subjects were not included in the analysis? Please clarify.
